# Correlates of Levels of Willingness to Engage in Climate Change Actions in the United States

**DOI:** 10.3390/ijerph18179204

**Published:** 2021-08-31

**Authors:** Carl A. Latkin, Lauren Dayton, Da-In Lee, Grace Yi, Mudia Uzzi

**Affiliations:** 1Department of Health, Behavior and Society, Johns Hopkins Bloomberg School of Public Health, Baltimore, MD 21205, USA; ldayton2@jhu.edu (L.D.); GraceYi@gmai.com (G.Y.); mudia.uzzi@jhmi.edu (M.U.); 2Division of Infectious Diseases, Johns Hopkins University School of Medicine, Baltimore, MD 21205, USA; 3Krieger School of Arts & Sciences, Johns Hopkins University, Baltimore, MD 21205, USA; dlee227@jhu.edu

**Keywords:** climate change actions, political ideology, environmentalism, peer influence, political party, activism

## Abstract

While the majority of the American public believe climate change is occurring and are worried, few are engaged in climate change action. In this study, we assessed factors associated with the level of willingness to engage in climate change actions using an online, longitudinal US study of adults. Climate change action outcomes included the level of willingness to post materials online, take political actions, talk with peers about climate change, and donate to or help an organization. Predictors included climate change attitudes, environmental attitudes, political ideology, political party affiliation, and demographic variables. Most (72%) of the 644 respondents only talked about climate change with peers a few times a year or less, though 65% were very or extremely worried about climate change. Many respondents indicated a willingness to do somewhat or a lot more, from 38% willing to talk to peers to 25% for willing to take political actions. In multinomial regression models, the Climate Change Concern scale was strongly and consistently associated with willingness to engage in climate change action. These findings indicate a need to both identify those who are willing to act and finding activities that fit with their interests and availability.

## 1. Introduction

A national poll conducted in December 2020 indicated that the majority of Americans are concerned about climate change, with 72% of respondents believing that global warming is occurring and 66% were “very” or “somewhat” worried about global warming [1]. However, prior research has found that climate change is not a frequent topic of conversation for most Americans [2]. Moreover, few Americans report that they are engaged in political activities about climate change, suggesting a disconnect between expressed concern and subsequent actions to address these concerns. The same December 2020 poll found that only 1% of Americans reported currently participating in a campaign to encourage elected officials to take action to reduce global warming. Conversely, more than a quarter of Americans polled said they would definitely (9%) or probably (20%) participate in such a campaign [3]. The discrepancy between the expressed level of concern about climate change and actual behaviors to address it suggests a need for a greater focus on providing the public with actionable options to engage in processes to address climate change. The focus of the paper was to examine the perceived potential for increased climate change activism among a sample of US adults. This national sample was drawn from an ongoing longitudinal study of well-being of Americans during the COVID-19 pandemic.

## 2. Literature Review

One explanation for the lack of public action on climate change is that, in some social circles, the label “environmentalist” carries negative connotations or stereotypes that may impede the affiliation with climate change activists [4]. Moreover, some people appear to view environmentalism as trendy or as a current generational “fad”. Research has also shown that opinions on climate change may vary based on social class [5], with some preferring to dedicate efforts and time to fulfill immediate economic concerns. As theorized by Laidley (2013), those with lower levels of cultural capital viewed individuals who are highly concerned about climate change as materially secure [5]. To the relatively poorer or less educated, the issue was perceived as a “rich person’s” concern, highlighting a social distinction in perceptions of climate change and suggesting that the issue has taken on greater symbolic meaning.

Attitudes toward climate change in the US have fluctuated over time, based on natural disasters, historical events, funding for political and business organizations, and increasing climate change skepticism that has been reinforced by certain news media outlets. Concern about climate change in the US increased after the US west coast fire of 2019, with 73% of sampled voters reporting that weather events such as wildfires increased their concerns about the impacts of climate change [1]. A poll by the Pew Foundation in 2020 found that 60% of respondents viewed climate change as a major threat to the well-being of the US, the highest rating since the poll began in 2009 [6]. Yet, the Trump administration’s systematic decimation of climate change policy ensured that public opinion was not translated into policy on a federal governmental level. The anti-climate actions by the Trump administration may have been driven in part by the widening partisan gap regarding climate change opinions.

Indeed, a large body of literature suggests that climate change attitudes are strongly linked to political partisanship and ideology. In the US, this manifests as significantly lower levels of concern about climate change among conservatives as compared to liberals, and, likewise, among Republicans as compared to Democrats and Independents. These widening partisan differences have been documented over time by Gallup polling. From 2001 to 2021, participants were asked the question: “Do you think that global warming will pose a serious threat to you or your way of life in your lifetime”? In 2001, there was a 16-point divide in affirmative answers between Republicans and Democrats; by 2016, there was a 38-point difference [7]. In 2021, this difference was 56 points, with 67% of Democrats considering global warming to be a serious threat to themselves, compared to 43% of Independents and 11% of Republicans. The climate change inaction among past Republican administrations highlights the importance of consistent political advocacy from the bottom up, as public opinion may not translate into policy. This partisan divide further suggests that working to involve Republicans in climate change action activities is critical.

Although there is a wealth of literature about the level of concern about climate change in the population, there is less information about willingness to engage in climate change actions. In the current paper, we assessed psychosocial and political factors associated with willingness to take direct political actions, indirect actions of helping an organization that addresses climate change, and actions that would change public opinion and increase public discussion of climate change.

We first examined the degree to which respondents expressed interest, if trained, in talking to family and friends about climate change, engaging in political actions such as calling a legislator, or attending a meeting about climate change. We also assessed participants’ willingness to donate to or help out an organization that addresses climate change and willingness to post materials online about climate change. We examined differences between expressed concern about climate change and willingness to engage in climate change action. We also examined the correlates of willingness to engage in climate change action. We further assessed whether political ideology and political party affiliation provided any additional explanatory power in identifying correlates of willingness to engage in climate change action. We hypothesized that (1) high concerns about climate change would predict willingness to engage in climate change action; (2) negative attitudes toward environmentalist and environmentalism would also predict willingness to engage in climate change action; and (3) political ideology and party affiliation would independently predict willingness to engage in climate change action. Given the political partisanship in climate change perspectives, we also assessed if there were factors unique to conservatives that predict their willingness and interest in participating in climate change actions. We hypothesize that limiting the sample to conservatives would reveal similar associations as what was found in the whole sample.

## 3. Materials and Methods

Study respondents were from an online longitudinal study that began in March 2020 during the COVID-19 pandemic. This study aimed to examine individual, social, and societal-level fluctuations amid the pandemic’s rapidly changing landscape. Study participants were recruited through Amazon’s Mechanical Turk (MTurk) service. This approach is regularly used by health researchers, as it allows for a diverse sample to be collected in a rapid and timely fashion [8]. Prior research has indicated that MTurk provides better quality data in less time than other methods for recruiting convenience samples [9]. Study populations recruited through MTurk are not nationally representative but have been documented to outperform other opinion samples on several dimensions [10]. Studies using MTurk have also demonstrated good reliability [11]. Study protocols followed MTurk’s best practices for research, which included ensuring participant confidentiality, protecting study integrity, generating unique completion codes, integrating validity-checks throughout the survey (which included low probability events such as frequency of deep-sea diving in Alaska and number of appendages that had been removed) and repeating study-specific qualification questions [9,12,13]. The demographic characteristics of MTurk appear to be stable [14], and although MTurk respondents are more liberal than the general public, those who are conservative do not significantly differ in their attitudes from conservatives recruited from other sources [15].

The key variables for this study were collected in the fifth wave of data collection, except for the variables of political ideology and political party affiliation, which were collected on the first wave. All respondents who successfully completed the first survey were invited to participate in the subsequent rounds of data collection. Study respondents completed the first survey between 24 and 27 March 2020. For the fifth wave of data collection, which was administered between 4 and 15 March 2021. During the fifth wave, the sample was replenished by enrolling an additional 94 participants for a sample of 644. Since MTurk samples tend to under-represent some minority populations, we specifically recruited African Americans and Latinx respondents into the fifth wave of study in order for the study to be more representative of the US population.

Eligibility for the study included being age 18 or older, living in the United States, being able to speak and read English, having heard of the coronavirus or COVID-19, and providing written informed consent. To enhance study validity, eligible participants had to pass attention and validity checks embedded in the survey. Participants were paid USD 4.25 for the fifth wave, which was equivalent to approximately USD 12 per hour. The study protocols were approved by the Johns Hopkins Bloomberg School of Public Health Institutional Review Board.

## 4. Measures

To assess willingness to become involved in climate change action activities, respondents were asked, “If trained, how much would you be willing to do the following activities?”: (1) talk to family and friends about climate change, (2) engage in political actions such as calling a legislator or attending a meeting about climate change, (3) donate to or help out an organization that addresses climate change, and (4) post materials online or emailing about climate change. The response options were “Not at all”, “A little more”, “Somewhat more”, and “A lot more”.

To assess attitudes toward environmentalists and climate change activists, we created a scale using a modified version of the Environmentalist Threat Measure, which consisted of eight items [16]. We included language about climate change, as the original scale only focused on environmentalism. We anticipated that the scale, which we refer to as the Attitudes Towards Environmentalists Scale, would measure a single manifest factor. Examples of items included: “Environmentalists are out of touch with the needs of most Americans”, “People who worry about climate change are often out of touch with the economic realities of most Americans”, and “If the American government makes changes to protect the environment, other countries will continue to pollute and get an economic advantage.” The response categories were “Strongly agree”, “Agree”, “Neither agree nor disagree”, “Disagree”, and “Strongly disagree”. A principal component analysis was conducted to assess the psychometric properties of the scale, which revealed that one factor accounted for 65.9 percent of the variance. The Cronbach’s alpha was 0.93.

To measure belief in global warming and motivation to address climate change [17], we use the Yale Program on Climate Change Communication’s Short Climate Survey [18], which has been used to subdivide individuals into six groups for audience segmentation [19]. The brief survey includes the following four items: (1) “How important is the issue of global warming to you personally?” (2) “How worried are you about global warming?” (3) “How much do you think global warming will harm you personally?” and (4) “How much do you think global warming will harm future generations of people?” Many prior analyses found a linear relationship between scores on these four items and outcomes. We used this measure as a scale, referred to as the Climate Change Concern Scale, as a principal component analysis revealed that one factor accounted for 82.5 percent of the variance and we calculated a Cronbach’s alpha of 0.93.

Political ideology was assessed with the question: “Where would you place yourself on a scale running from “Very liberal” to “Very conservative?” The response categories were “Very Liberal (7)”, Liberal (6)”, Slightly Liberal (5)”, “Moderate (4)”, “Slightly Conservative (3)”, “Conservative (2)”, and “Very Conservative (1)”. There were three individuals who did not provide data to this question and were assigned a median value of 3. Political party affiliation was assessed with the standard question, “Do you consider yourself Republican, Democrat, Independent, Libertarian or other?” Due to the small cell size, the Libertarian and the “Other” groups were collapsed into the “Other” group.

Gender, education, and income were also assessed. Level of education was collapsed to reflect some college or less versus bachelor’s degree or higher. Income was collected as a categorical variable and then dichotomized at the median of USD 60,000 or below. Two individuals with missing income data were assigned the median value and included in the median and below group. The response categories for self-reported race/ethnicity included “White”, “Black”, “Asian”, “Hispanic”, “Mixed”, or “Other”. Due to the sample size, the categories of “Asian”, “Mixed”, and “Other” were collapsed.

## 5. Analysis

Descriptive statistics of means, standard deviations, and percentages were first calculated. Bivariate and multinomial logistic regression models were employed to assess the differences in increased willingness, if trained, to engage in the four types of climate change action: (1) talking to family and friends, (2) calling legislators or attending meetings, (3) donating to or helping out an organization, and (4) posting materials online or email. The three groups were (1) A little more, (2) Somewhat more, and (3) A lot more. The reference group was “Not at all more”. Multinominal logistic regression models were selected since the dependent variable was categorical to assess the independent contribution of each of the predictor variables. The multivariable models adjusted for income and education.

As concern about climate change has become increasingly normalized, we felt that social desirability biases might incline more respondents to express interest in addressing the issue. Hence, we thought an ordinal measure of the intensity of prospective engagement would provide a more nuanced perspective. This approach also allowed us to examine the linearity of the association between the covariates and the intensity of willingness to engage in climate change actions.

In considering whether to use the political ideology or party affiliation variable as a covariate in the multinomial bivariate and multivariate regression models, we observed that there was no significant difference between the models produced by the variables of political ideology or party affiliation. For the bivariate models, the reported coefficients of the independent group (party affiliation) for each outcome were within the 95% confidence intervals of the coefficients of political ideology as measured by the political ideology sliding scale (very liberal to very conservative). The coefficients of the Republican and Libertarian/Other categories were lower, but this may be due to the smaller group sizes. For the multivariate models, the coefficients of the Republican and independent groups were within the 95% confidence intervals of the political ideology sliding scale, while the coefficients of Libertarian/Other categories were lower, again due to its small sample size. In general, the confidence intervals for each of these groups were larger than those models using the political ideology sliding scale, which indicates that the political ideology sliding scale may be a more accurate representation of political views and thus the better feature to include in our models. We also conducted an analysis restricting the models to conservatives to examine whether there were different associations among this group with the outcomes. These results (Data not shown) did not reveal substantial differences and hence only the full sample findings are presented. A post hoc analysis also revealed that the Attitudes Towards Environmentalists Scale and the Climate Change Concern Scale were highly correlated. In regression models, covariates representing these two scales used z-score values.

## 6. Results

Sample demographics (N = 644) are reported in Table 1. The mean age of the sample was 40.3 (SD 11.6). More than half (68.3%) of the sample was White, while 13.8% of participants were non-Hispanic Black race/ethnicity, 8.4% were Hispanic, 6.1% were Asian, and 3.4% reported Mixed/Other race. Half of respondents reported female gender (55.8%), an income of USD 60,000 or less (56.1%), and an educational attainment of a Bachelor’s degree or higher (56.8%). Just under half of the sample identified as Democrats (48.6%), with 18.8% as Republicans, 27.2% as Independents, and the remaining as Libertarians (2.8%) or Other (2.6%).

Survey results using the Climate Change Concern Scale are reported in Table 1 and revealed that a greater number of individuals expressed concern about the impact of climate change than were willing to engage in “a lot more” action to address it. More than half of the sample (66.0%) reported that global warming was extremely or very important to them personally, and 65% were extremely, very or worried about global warming. Moreover, the majority of the sample (66.8%) felt that global warming would personally harm them a great deal or a moderate amount, while most (91.7%) believed that global warming would harm future generations a great deal or moderate amount. However, the vast majority of the sample (72.2%) reported talking about global warming to family and friends only a few times a year or never.

Overall, willingness to engage in “a lot more” activity ranged from 7% for political actions to 13% for talking to family and friends. In general, the group that reported that they would be willing to engage “somewhat more” in climate change action was two to three times as large as the group that reported that they were willing to engage in “a lot more” action. As the sample was over-representative of Democrats, we reported willingness to engage in four different climate change actions stratified by political party (Table 2). Compared to the overall average, Republicans were less likely to endorse that they were willing to engage in climate change actions a little, somewhat, or a lot more such as calling legislators (53.7% overall, 29.8% for Republicans), posting materials online (57.6% overall, 34.7% for Republicans) and donating to or helping out organizations that address climate change (63.3% overall, 39.7% for Republicans). Among Republicans, participants were most receptive to talking to friends and family about climate change compared to other climate change actions (49.6% a little, somewhat, or a lot more willing).

All the multivariable models adjust for income and education. Bivariable and multivariable models of willingness to talk to friends and family about climate change are reported, if trained, are displayed in Table 3. Increasing receptivity towards environmentalist’s attitudes, political affiliation (more liberal) and age (younger) were all positively associated with increased willingness to engage family and friends about climate change with training, though these associations were not attenuated in the multivariable model. However, scores on the Climate Change Concern Scale were significantly and independently associated with willingness to talk with family friends about climate change if trained, with higher scores significantly associated with engaging “a little more” (aOR = 3.8, 95% CI: 2.6–5.6), “somewhat more” (aOR = 7.6, 95% CI 4.7–12.3), and “a lot more” (aOR = 15.9, 95% CI: 8.3–30.7) compared to not at all.

As reported in Table 4, there were relatively more correlates of willingness to take political actions such as calling a legislator or attending a meeting about climate change. As with the previous model, scores on the Climate Change Concern Scale were independently associated with “a little more” (aOR = 2.7, 95% CI: 1.9–3.7), “somewhat more” (aOR = 3.4, 95% CI: 2.3–5.0), and “a lot more” willingness to take political action (aOR = 10.9, 95% CI: 4.9–24.2). Increasing receptivity to environmentalist’s attitudes was associated with all degrees of increasing willingness, though this association was only attenuated in multivariable models. Compared with White race, non-Hispanic Black race was independently associated with “a little more” willingness to take political actions (aOR = 2.2, 95% CI: 1.2–3.9). In addition, increasingly liberal political affiliation was independently associated with “somewhat more” willingness (aOR = 0.8, 95% CI: 0.7–1.0) to take political actions and female sex (aOR = 2.2, 95% CI: 1.0–4.7) was associated with “a lot more” willingness.

A similar pattern of association emerged for models on willingness to donate or help out an organization that addresses climate change (Table 5). As with both previous models, the Climate Change Concern Scale was independently and significantly associated with “a little more”, “somewhat more”, and “a lot more” willingness to donate or help out. In bivariable models, receptivity to environmentalist’s attitudes was associated with all levels of increasing willingness, though none remained significant in multivariable models. Compared to white participants, non-Hispanic Black race was independently associated with “a little more” (aOR = 2.6, 95% CI: 1.4–5.0) and “somewhat more” willingness (aOR = 3.0, 95% CI: 1.5–6.3) to help out or donate. Similarly, Asian/Mixed/Other participants were “a lot more” willing to do so (aOR = 4.4, 95% CI: 1.6–12.3) compared to white participants. As with willingness to take political actions such as calling legislators, females were “a lot more” willing to help out or donate (aOR = 3.9, 95% CI: 1.8–8.4). Political affiliation and age were associated in bivariable models, but these associations were attenuated in multivariable models.

Table 6 reports correlates of willingness to post materials online or e-mail about climate change. Again, scores on the Climate Change Concern Scale were significantly associated with all levels of increasing willingness (“A little more” aOR = 2.4, 95% CI: 1.7–3.4, “Somewhat more” aOR = 5.6, 95% CI: 3.6–8.6, “A lot more” aOR = 9.2, 95% CI: 4.8–17.7). Hispanic race, compared to white, was independently associated with “a little more” willingness to post materials online or email (aOR = 3.1, 95% CI: 1.0–9.0). Increasingly liberal affiliation was also independently associated with “a little more” willingness to post materials or email (aOR = 0.8, 95% CI: 0.7–1.0).

Overall, scores on the Climate Change Concern Scale were strong and consistent predictors of willingness to engage in climate change action. These scale items were originally developed to categorize respondents into six groups with diverse climate change attitudes and behaviors, ranging from alarmists to denialists. However, as previously discussed, the items also manifest strong scale properties. It is not surprising that those who are more concerned about the impact of climate change are more likely to report willingness to engage in climate change actions. The Chi-square for the Likelihood Test Ratio for the scale ranged from 82 to 133, whereas the highest Chi-square for any of the other variables was 23 (data not shown). These findings suggest that a brief measure can readily identify those who express a strong willingness to engage in climate change action. It may be useful to examine how individuals’ social media postings may be associated with scores on the Climate Change Concern Scale in order to identify individuals who may be willing to engage in more climate change actions.

Finally, it is interesting to note that political ideology did not appear to add predictive power to most of the models. Political ideology was statistically significant in 11 of the 12 bivariate models but remained significant in only 2 out of 12 of the multivariable models. In multivariable models, more liberal political ideology was associated with being “somewhat more” willing to engage in political actions such as calling a legislator or attending a meeting about climate change (aOR = 0.8, 95% CI = 0.7–1.0) compared to being “not at all more” willing to engage. In addition, more liberal ideology was associated and independently with being “a little more” willing to post materials online or e-mail about climate change (aOR = 0.8, 95% CI = 0.7–1.0). Significant bivariate findings were not attenuated in the rest of the multivariable models.

## 7. Discussion

We hypothesized that (1) high concerns about climate change would predict willingness to engage in climate change action; (2) negative attitudes toward environmentalist and environmentalism would also predict willingness to engage in climate change action; and (3) political ideology and party affiliation would independently predict willingness to engage in climate change action. In this study, multivariable models revealed that the only consistent and strong predictor of willingness to increase the amount of climate change activism was the level of concern about climate change from the Climate Change Concern Scale. This association provides additional validation to the measure. Although willingness to engage in climate change actions was associated with political ideology in 11 of the 12 bivariate models, it was only significantly associated with climate change actions in 2 of the 12 models. On the other hand, the Climate Change Concern Scale was significant in all the bivariate and multivariable models. The direct and consistent association between scores on the Climate Change Concern Scale and willingness to engage in multiple dimensions of climate change advocacy suggests that a brief Climate Change Concern 4-item survey may be an efficient and instrumental tool in identifying potential climate change activists, independent of political ideology. In limiting the models to conservatives, there were no noteworthy changes in the findings.

There were some notable sociodemographic differences. Hispanic respondents, compared to whites, were much more willing to post materials online. Black respondents were “somewhat more willing” than white individuals to donate to or help out such an organization. Compared to men, women were a lot more willing to engage in political actions such as calling a legislator or attending a meeting about climate change. Gender differences in concern about climate change have been noted in prior studies [1,7]. There was also an association between greater political conservativism and being “a little” less likely to be willing to post materials online about climate change or engage in “somewhat more” political action, which is also consistent with prior research [1,6,7].

The increased willingness of the survey respondents of color to engage in climate change action may be related to the specific impact climate change has in their communities. People of color are more vulnerable to environmental hazards and other harmful effects of climate change [20,21]. Much of this is due to environmental racism structuring the geographic location of predominately Black, Brown, Asian, and Indigenous neighborhoods as well as the built environment within these neighborhoods [22,23,24]. Climate change leads to frequent extreme weather events, which have a disproportionate impact on Black, Brown, and Indigenous neighborhoods [25,26]. For example, majority Black neighborhoods in Texas suffered a greater extent from flooding in comparison to other neighborhoods after Hurricane Harvey in 2017 [27]. Neighborhoods of color are also more likely to be susceptible to climate change with one study finding that formerly redlined neighborhoods, mostly inhabited by people of color, have, on average, 23% less tree canopy cover than other neighborhoods [28]. The lack of tree canopy cover in these neighborhoods puts people of color at increased risk of heat-related deaths due to the formation of heat islands within urban communities [26]. Moreover, communities of color are more vulnerable to the social and economic consequences of climate change in the form of climate gentrification [29]. Ten years after Hurricane Katrina, gentrification was strongly associated with hurricane-impact in New Orleans with census tracts on higher ground elevation [30] more likely to be gentrified. Researchers observed a similar phenomenon of flooding and gentrification of Black and Brown communities with higher ground elevation in Miami [31]. While women and people of color have been underrepresented in high-level climate change negotiations [32,33], they have been heavily engaged in climate change action on the ground level. More resources and investment need to be provided to historically marginalized groups to lead on climate change action. Moreover, it is important to understand the unique needs and barriers of underrepresented groups to ensure that barriers to sustained involvement in climate change action, especially in leadership positions, are addressed.

These data also suggest that there is a large number of US residents who are willing to engage in climate change action. The proportion of respondents who reported willingness to engage in “somewhat more” and “a lot more” in the four climate change action activities ranged from 18 to 27%. Among Republicans, willingness to engage “somewhat more” and “a lot more” in the four climate change action activities ranged from 14 to 17%. Although these numbers are lower than the total sample, they still represent a large number of people who may be willing to engage in climate change action. Even a small increase in the percentage of Republicans who engaged in climate change actions through lobbying Republican legislators to pass meaningful legislation could have a major impact on policymakers.

Given the sharp partisan divide over the climate change emergency in the US, one challenge is to understand how to engage conservatives who may receive news and information from sources that downplay climate change. As environmentalism is frequently affiliated with liberal groups, it is important to ensure that conservatives are provided support and relevant information to promote climate action. It may be beneficial for organizations to develop strategies to identify conservatives who might be more willing to engage in climate change action. These strategies may benefit by including domains typically rated as being important to conservatives, such as family and economic freedom. For example, Gustafson et al. (2020) developed messages that highlighted how climate change is impacting fishing and hunting in the Southern US [34]. These researchers found that conservatives were receptive to such messages. Messages could also emphasize the need for the recipient to show leadership and act in a way that is perceived as “right for the country”—even if such actions are not comfortable.

Increasing the diversity of participants in climate change organizations may also enhance diversity in advocacy tactics, as different individuals and organizations may have different forms of campaigns. Greater diversity also reduces the ability of oppositional groups to isolate climate action activists and label them as extremists [35,36]. In an analysis of a failed cap-and-trade bill in the US congress, Skocpol (2013) suggests that environmental organizations should have focused less on mobilizing support within their organizations that have similar types of members, and instead working with more diverse groups such as business sectors to secure a deal [37]. Increasing diversity of political and religious ideologies within organizations, for example, could enhance climate change action.

The gap between climate change attitudes and behaviors, which has been found in other studies, was replicated here with a reported high level of concern about climate change but it being an infrequent topic of conversation. There is also a gap between reported level of concern and climate action [3]. From a behavioral change perspective, a range of approaches to increase the strength of association between attitudes and behaviors has been previously identified; these include reducing the cost of actions, cueing actions, making behaviors into habits, heightening social norms, and integrating actions into social identities [38]. From a policy and public health perspective, changing the behaviors of a relatively small percentage of the population can lead to substantial cultural change. For substantive climate change action, lobbying legislators may require fewer individuals than changing public opinion through the diffusion of messaging (through family and friends or posting materials on climate change). All of these approaches are likely to be synergistic.

Taken together, data from this and previous studies suggest that there are high levels of expressed willingness to engage in climate change action. To the extent that a high level of willingness to engage in climate change action might be an accurate reflection of *potential* behaviors, it is critical to understand how to translate willingness into action. Future programs and research should assess the nature of activities and engagement that are rewarding, sustainable, and effective, and also delineate varying levels of time commitment that individuals are willing to give. With numerous competing demands on legislators’ time, climate change actions need to maintain salience and influence. One approach to gauging potential actions is to ask individuals what they are willing to do and what would help them achieve these goals in terms of support, skills, resources, and opportunities. Goldberg et al. (2020) outline several strategies to promote behavior change to address climate change [39].

One approach to increase climate change action among those who are willing to do “a little more” is to frame messages so climate actions do not seem overly burdensome. Engaging in climate change actions may promote an individual or social identity based on addressing climate change, which can lead to additional climate change actions [40]. Prior research also suggests that engagement in smaller actions can lead to greater investments of time and resources in addressing social and political issues [41]. However, it is necessary to consider that condensing climate change advocacy to more “private” actions, such as quickly sending an email to a legislator, may have the unanticipated consequences of not heightening social norms, modeling behaviors, or obtaining social rewards [42]. Thus, it may be advantageous to link these actions to the formation of small groups that provide support. A complementary approach might be to train individuals who report that they are willing to do “a lot more” about climate change to become community leaders. One such example is the #makeitbetter campaign [43] by OPHA, which encouraged organizing at a neighborhood level as well as engaging in community initiatives or participating in policy discussions at the provincial level. Such guidance also needs to include skill training. This approach may be especially beneficial in addressing climate change among conservatives, as there is a paucity of climate change leadership among this group.

The threat of climate change is unprecedented in both scope and magnitude. Major national and international policy changes are necessary in order to have meaningful reductions in climate change. Consequently, the approaches to addressing it must be greater and engage more diverse advocates. The study results highlight the untapped potential for climate change action. Although this was not a random sample, these data suggest that there is a large number of people who are willing to engage in increased climate change action. If 1% of the US population is currently involved in lobbying legislators about climate change, doubling this number would yield a significant number of people involved in climate change action. Given its scope, the formidable task of tackling climate change necessitates the involvement of numerous individuals. To engage more people in climate change actions, we need to assess what people are willing to do to address climate actions and what actions will lead to meaningful and impactful policy changes. There is also a need to reward and support those who are engaged in climate change actions.

## 8. Study Limitations

The study limitations should be noted. The sample over-represented Democrats and was not random, hence limiting generalizability. However, prior MTurk studies have found that Republicans recruited by MTurk respond similarly to other samples of Republicans. To address this sampling bias, we stratified many of the analyses by political party and political ideology. We did not collect information on the current level of climate change activism among the sample. However, this and prior studies have found a low frequency of even talking about climate change. We also do not know if professed interest in addressing climate change and training opportunities would lead to subsequent climate change actions.

## 9. Conclusions

The study findings suggest that regardless of political affiliations, age, race, and gender, individuals who express concern about climate change also express willingness to talk to others about climate change. We also found that those who express concern about climate change and women compared to men were much more likely to indicate that they would do a lot more in terms of political actions or donations to organizations that address climate change. A critical issue for future research and praxis is turning willingness into action. Future research should focus on the high willingness group to identify barriers and motivators to engage in climate change action.

## Figures and Tables

**Table 1 ijerph-18-09204-t001:** Background factors and climate change attitudes among respondents (N = 644).

		Mean (SD)	Percentages
Age		40.30 (11.62)	
Race			White: 68.27% Non-Hispanic Black: 13.84% Hispanic: 8.40% Asian: 6.07% Mixed/Other: 3.42%
Biological Sex			Male: 44.17% Female: 55.83%
Income			USD 60,000 or less: 56.14% Greater than $60,000: 43.86%
Education			Some college or less: 43.23% Bachelors degree or higher: 56.77%
Political Party Affiliation			Republican: 18.78% Democrat: 48.60% Independent: 27.17% Libertarian: 2.80% Other: 2.64%
Political Ideology		3.29 (1.76)	Very liberal: 14.15% Liberal: 29.39% Slightly liberal: 14.46% Moderate: 18.35% Slightly conservative: 7.62% Conservative: 10.58% Very conservative: 5.44%
Climate Change Concern Scale items	“How important is the issue of global warming to you personally?”	3.86 (1.05)	Not at all important: 3.26% Not too important: 6.52% Somewhat important: 24.24% Very important: 33.40% Extremely important: 32.59%
	“How worried are you about global warming?”	2.87 (0.92)	Not worried at all: 7.13% Somewhat worried: 28.31% Very worried: 34.62% Extremely worried: 29.94%
	“How much do you think global warming will harm you personally?”	2.78 (0.80)	Not at all: 5.91% Only a little: 27.29% A moderate amount: 49.29% A great deal: 17.52%
	“How much do you think global warming will harm future generations of people?”	3.57 (0.72)	Not at all: 2.65% Only a little: 5.70% A moderate amount: 23.83% A great deal: 67.82%
	Climate Change Concern Scale Total	12.06 (3.75)	
Attitudes Towards Environmentalists Scale Total		24.91 (7.06)	
How often do you talk to your friends and family about climate change?			Never: 27.80% A few times a year: 44.41% Several times a month: 24.22% Every week: 3.57%

**Table 2 ijerph-18-09204-t002:** Willingness to engage in climate change action by political party affiliation (N = 644).

	If Trained, How Much Would You Be Willing to Do the Following Activities?				
	**Talking to friends and family about climate change**				
	**Not at all**	**A little**	**Somewhat**	**A lot more**	**Total**
**Democrat**	41 (13.10%)	120 (38.34%)	100 (31.95%)	52 (16.61%)	313
**Republican**	61 (50.41%)	39 (32.23%)	18 (14.88%)	3 (2.48%)	121
**Independent**	52 (29.71%)	65 (37.14%)	37 (21.14%)	21 (12.00%)	175
**Libertarian/Other**	15 (42.86%)	10 (28.57%)	5 (14.29%)	5 (14.29%)	35
**Total**	169 (26.24 %)	234 (36.34%)	160 (24.84%)	81 (12.58%)	644
	**Political actions such as calling a legislator or attending a meeting about climate change**				
	**Not at all**	**A little**	**Somewhat**	**A lot more**	**Total**
**Democrat**	100 (31.95%)	103 (32.91%)	79 (25.24%)	31 (9.90%)	313
**Republican**	85 (70.25%)	19 (15.70%)	13 (10.74%)	4 (3.31%)	121
**Independent**	86 (49.14%)	56 (32.00%)	23 (13.14%)	10 (5.71%)	175
**Libertarian/Other**	27 (77.14%)	6 (17.14%)	2 (5.71%)	0 (0.00%)	35
**Total**	298 (46.27%)	184 (28.57%)	117 (18.17%)	45 (6.99%)	644
	**Donating to or helping out an organization that addresses climate change**				
	**Not at all**	**A little**	**Somewhat**	**A lot more**	**Total**
**Democrat**	77 (24.60%)	104 (33.23%)	99 (31.63%)	33 (10.54%)	313
**Republican**	73 (60.33%)	30 (24.79%)	14 (11.57%)	4 (3.31%)	121
**Independent**	66 (37.31%)	56 (32.00%)	40 (22.86%)	13 (7.43%)	175
**Libertarian/Other**	20 (57.14%)	11 (31.43%)	3 (8.57%)	1 (2.86%)	35
**Total**	236 (36.65%)	201 (31.21%)	156 (24.22%)	51 (7.92%)	644
	**Posting materials online or emailing about climate change**				
	**Not at all**	**A little**	**Somewhat**	**A lot more**	**Total**
**Democrat**	89 (28.43%)	107 (34.19%)	82 (26.20%)	35 (11.18%)	313
**Republican**	79 (65.29%)	21 (17.36%)	15 (12.40%)	6 (4.96%)	121
**Independent**	83 (47.43%)	50 (28.57%)	28 (16.00%)	14 (8.00%)	175
**Libertarian/Other**	22 (62.86%)	8 (22.86%)	1 (2.86%)	4 (11.43%)	35
**Total**	273 (42.39%)	186 (28.88%)	126 (19.57%)	59 (9.16%)	644

**Table 3 ijerph-18-09204-t003:** Bivariable and multivariable model of willingness to talk to friends and family about climate change, if trained.

	If Trained, How Much Would You Be Willing to Do the Following Activities? Talking to Friends and Family about Climate Change (N = 644) (REF: Not at All)					
	A little more		Somewhat more		A lot more	
	OR (95% CI)	aOR (95% CI)	OR (95% CI)	aOR (95% CI)	OR (95% CI)	aOR (95% CI)
Climate Change Concern Scale	**3.92** **(2.97–5.16)**	**3.81** **(2.57–5.64)**	**8.54** **(5.88–12.40)**	**7.61** **(4.73–12.26)**	**21.49** **(12.34–37.42)**	**15.92** **(8.25–30.71)**
Attitudes Towards Environmentalists Scale	**2.40** **(1.91–3.01)**	0.87 (0.60–1.27)	**3.82** **(2.87–5.09)**	1.04 (0.67–1.61)	**7.71** **(4.99–11.91)**	1.59 (0.88–2.87)
Race–Non-Hispanic Black (REF: White)	1.53 (0.83–2.84)	1.39 (0.68–2.86)	1.72 (0.88–3.36)	1.91 (0.85–4.32)	1.82 (0.85–3.93)	2.45 (0.94–6.40)
Race–Hispanic (REF: White)	1.54 (0.69–3.43)	0.94 (0.37–2.38)	**3.09** **(1.41–6.80)**	1.91 (0.72–5.06)	0.47 (0.10–2.21)	0.28 (0.05–1.57)
Race–Asian/ Mixed/Other (REF: White)	1.62 (0.76–3.47)	1.15 (0.47–2.77)	2.20 (0.99–4.87)	1.47 (0.56–3.89)	2.13 (0.86–5.31)	1.55 (0.50–4.84)
Political Affiliation	**0.64** **(0.56–0.72)**	0.87 (0.738–1.03)	**0.56** **(0.48–0.64)**	0.90 (0.74–1.09)	**0.46** **(0.38–0.57)**	0.88 (0.69–1.14)
Age	**0.98** **(0.96–0.99)**	1.00 (0.97–1.02)	**0.98** **(0.96–1.00)**	1.00 (0.97–1.02)	0.98 (0.96–1.01)	0.99 (0.96–1.02)
Sex (REF: Male)	1.26 (0.85–1.88)	1.17 (0.72–1.90)	1.32 (0.86–2.04)	1.26 (0.73–220)	1.58 (0.92–2.70)	1.61 (0.82–3.18)

Bold: *p* ≤ 0.05.

**Table 4 ijerph-18-09204-t004:** Bivariable and multivariable model of willingness to call legislator or attend a meeting about climate change, if trained.

	If Trained, How Much Would You Be Willing to Do the Following Activities? Political Actions Such as Calling a Legislator or Attending a Meeting about Climate Change (N = 644) (REF: Not at All)					
	A Little More		Somewhat More		A Lot More	
	OR (95% CI)	aOR (95% CI)	OR (95% CI)	aOR (95% CI)	OR (95% CI)	aOR (95% CI)
Climate Change Concern Scale	**2.85** **(2.23–3.64)**	**2.67** **(1.91–3.73)**	**3.10** **(2.32–4.16)**	**3.39** **(2.30–4.99)**	**12.87** **(6.43–25.74)**	**10.92** **(4.93–24.19)**
Attitudes Towards Environmentalists Scale	**2.04** **(1.65–2.53)**	0.96 (0.69–1.35)	**1.89** **(1.48–2.41)**	**0.68** **(0.47–0.99)**	**4.81** **(2.88–8.01)**	1.07 (0.57–2.01)
Race–Non-Hispanic Black (REF: White)	**2.14** **(1.25–3.64)**	**2.16** **(1.20–3.91)**	1.52 (0.81–2.86)	1.64 (0.82–3.26)	1.14 (0.41–3.15)	1.73 (0.55–5.40)
Race–Hispanic (REF: White)	**2.46** **(1.26–4.77)**	1.87 (0.90–3.89)	1.50 (0.67–3.39)	1.21 (0.50–2.93)	1.62 (0.51–5.12)	1.95 (0.54–7.07)
Race–Asian/ Mixed/Other (REF: White)	1.36 (0.73–2.55)	0.91 (0.46–1.82)	0.77 (0.34–1.77)	0.60 (0.25–1.46)	1.56 (0.60–4.09)	1.63 (0.53–5.00)
Political Affiliation	**0.69** **(0.61–0.78)**	0.90 (0.77–1.05)	**0.66** **(0.57–0.76)**	**0.80** **(0.67–0.95)**	**0.51** **(0.40–0.66)**	0.78 (0.58–1.04)
Age	0.98 (0.96–0.99)	0.99 (0.97–1.01)	1.00 (0.98–1.02)	1.02 (0.99–1.04)	1.01 (0.98–1.04)	1.03 (0.99–1.06)
Sex (REF: Male)	1.27 (0.88–1.85)	1.27 (0.84–1.92)	1.19 (0.77–1.82)	1.08 (0.68–1.74)	**2.03** **(1.04–3.97)**	**2.19** **(1.03–4.65)**

Bold: *p* ≤ 0.05.

**Table 5 ijerph-18-09204-t005:** Bivariable and multivariable model of willingness to donate or help out an organization that addresses climate change, if trained.

	If Trained, How Much Would You Be Willing to Do the Following Activities? Donating to or Helping out an Organization That Addresses Climate Change (N = 644) (REF: Not at All)					
	A Little More		Somewhat More		A Lot More	
	OR (95% CI)	aOR (95% CI)	OR (95% CI)	aOR (95% CI)	OR (95% CI)	aOR (95% CI)
Climate Change Concern Scale	**2.63** **(2.09–3.33)**	**2.39** **(1.70–3.35)**	**6.11** **(4.36–8.58)**	**6.30** **(4.10–9.68)**	**11.07** **(6.12–20.04)**	**10.23** **(5.05–20.73)**
Attitudes Towards Environmentalists Scale	**2.09** **(1.69–2.58)**	1.14 (0.81–1.60)	**2.84** **(2.20–3.67)**	0.90 (0.61–1.32)	**4.35** **(2.78–6.81)**	1.16 (0.65–2.09)
Race–Non-Hispanic Black (REF: White)	**2.34** **(1.30–4.23)**	**2.61** **(1.37–4.99)**	**2.32** **(1.26–4.29)**	**3.03** **(1.46–6.27)**	1.78 (0.66–4.79)	2.93 (0.96–8.92)
Race–Hispanic (REF: White)	**2.32** **(1.17–4.59)**	1.88 (0.89–4.01)	1.16 (0.50–2.68)	0.81 (0.31–2.08)	2.08 (0.70–6.16)	2.03 (0.59–6.99)
Race–Asian/ Mixed/Other (REF: White)	1.45 (0.73–2.89)	1.78 (0.55–2.52)	1.35 (0.65–2.83)	1.03 (0.43–2.43)	**3.81** **(1.64–8.89)**	**4.37** **(1.55–12.33)**
Political Affiliation	**0.73** **(0.66–0.82)**	0.99 (0.85–1.16)	**0.61** **(0.54–0.70)**	0.92 (0.77–1.197)	0.56 (0.45–0.70)	0.90 (0.69–1.18)
Age	**0.98** **(0.97–1.00)**	1.00 (0.98–1.02)	0.99 (0.97–1.00)	1.00 (0.98–1.02)	1.00 (0.97–1.02)	1.02 (0.98–1.05)
Sex (REF: Male)	1.45 (0.99–2.12)	1.46 (0.95–2.23)	1.39 (0.92–2.08)	1.42 (0.87–2.31)	**3.05** **(1.55–6.02)**	**3.88** **(1.79–8.42)**

Bold: *p* ≤ 0.05.

**Table 6 ijerph-18-09204-t006:** Bivariable and multivariable model of willingness to post materials or e-mail about climate change, if trained.

	If Trained, How Much Would You Be Willing to Do the Following Activities? Posting Materials Online or Emailing about Climate Change (N = 644) (REF: Not at All)					
	A Little More		Somewhat More		A Lot More	
	OR (95% CI)	aOR (95% CI)	OR (95% CI)	aOR (95% CI)	OR (95% CI)	aOR (95% CI)
Climate Change Concern Scale	**2.54** **(2.01–3.21)**	**2.43** **(1.73–3.42)**	**4.67** **(3.35–6.50)**	**5.58** **(3.63–8.58)**	**9.49** **(5.47–16.49)**	**9.22** **(4.80–17.71)**
Attitudes Towards Environmentalists Scale	**1.87** **(1.52–2.30)**	0.85 (0.60–1.20)	**2.14** **(1.67–2.74)**	0.69 (0.46–1.02)	**3.53** **(2.36–5.29)**	0.89 (0.61–1.53)
Race–Non-Hispanic Black (REF: White)	1.59 (0.91–2.78)	1.47 (0.80–2.70)	1.80 (0.99–3.27)	1.82 (0.92–3.61)	1.32 (0.54–3.22)	1.65 (0.61–4.50)
Race–Hispanic (REF: White)	**2.23** **(1.09–4.55)**	1.67 (0.77–3.63)	1.94 (0.87–4.33)	1.33 (0.54–3.28)	**3.11** **(1.23–7.88)**	**3.06** **(1.04–8.96)**
Race–Asian/ Mixed/Other (REF: White)	1.84 (0.98–3.44)	1.42 (0.71–2.83)	0.77 (0.32–1.88)	0.52 (0.20–1.37)	**2.38** **(1.02–5.60)**	2.28 (0.84–6.20)
Political Affiliation	**0.68** **(0.61–0.77)**	**0.84** **(0.72–0.98)**	**0.65** **(0.57–0.74)**	0.88 (0.74–1.06)	**0.57** **(0.46–0.69)**	0.84 (0.65–1.07)
Age	**0.97** **(0.96–0.99)**	0.99 (0.97–1.01)	**0.97** **(0.96–0.99)**	0.98 (0.96–1.01)	0.99 (0.97–1.02)	1.01 (0.98–1.04)
Sex (REF: Male)	1.11 (0.76–1.62)	1.13 (0.74–1.71)	1.10 (0.72–1.67)	1.05 (0.65–1.71)	**1.84** **(1.02–3.35)**	2.00 (1.02–3.82)

Bold: *p* ≤ 0.05.

## Data Availability

The data are available upon request from the first author.

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
