# Peer review of "Correlates of Levels of Willingness to Engage in Climate Change Actions in the United States"

_ijerph, 2021, doi:10.3390/ijerph18179204_

Round 1

Reviewer 1 Report

The evaluated manuscript is a resubmitted version of #ijerph-1255921. The authors responded to my suggestions and answered my questions. Thus, my recommendation is for the acceptance of it.

Author Response

Thank you for your thoughtful review

Reviewer 2 Report

This is an interesting paper which is aimed at assessing factors associated with the level of willingness to engage in climate change actions using an online, longitudinal US study of adults. Climate change action outcomes included the level of willingness to post materials online, take political actions, talk with peers about climate change, donate to or help an organization. Despite the noted strengthens, just a couple of major observations that would be needed to enhance the quality of the revised manuscript. These shortcomings are in the areas of literature review, research methods, discussions, and conclusions section. These are discussed as follows: However, the discussion is too lengthy and the conclusion section is not informative.

Literature review – Currently, this manuscript is devoid of a standalone ‘literature review’ section and the best practice normally is to have one that would then end up with a summary of the key issues and research or knowledge gap that needs to be addressed, and how this links to the purpose of the paper.

Research methods: The rationale for selection of the selection of the data analysis techniques not reported. Further, there is a mixture of ‘Harvard’ and ‘Numerical’ style of referencing within this ‘materials and methods’ section. Please revise accordingly. Examples of such Harvard types of ‘in-text’ references include the following: (Créquit et al., 2018); (Huff & Tingley, 2015); (Chandler & Shapiro, 2016); (Moss et al., 2020); (Follmer, Sperling, & Suen, 2017).

Please provide some supporting literature to the following statements: To enhance study validity, eligible participants had to pass attention and validity checks embedded in the survey.

Conclusion section – This need to be revisited through inclusion of the emergent contributions to knowledge as well as the limitations of the study should be reported. Currently, this section is very narrow.

Round 2

Reviewer 2 Report

The authors should be commended for the professional manner in which they have addressed the comments.